# NEURAL BREGMAN DIVERGENCES FOR DISTANCE LEARNING

**Fred Lu**[*†]**, Edward Raff**[*†]**, & Francis Ferraro**[*]
[*]University of Maryland, Baltimore County    [†]Booz Allen Hamilton
{lu_fred,raff_edward}@bah.com, ferraro@umbc.edu

## ABSTRACT

Many metric learning tasks, such as triplet learning, nearest neighbor retrieval, and visualization, are treated primarily as embedding tasks where the ultimate metric is some variant of the Euclidean distance (e.g., cosine or Mahalanobis), and the algorithm must learn to embed points into the pre-chosen space. The study of non-Euclidean geometries is often not explored, which we believe is due to a lack of tools for learning non-Euclidean measures of distance. Recent work has shown that Bregman divergences can be learned from data, opening a promising approach to learning asymmetric distances. We propose a new approach to learning arbitrary Bergman divergences in a differentiable manner via input convex neural networks and show that it overcomes significant limitations of previous works. We also demonstrate that our method more faithfully learns divergences over a set of both new and previously studied tasks, including asymmetric regression, ranking, and clustering. Our tests further extend to known asymmetric, but non-Bregman tasks, where our method still performs competitively despite misspecification, showing the general utility of our approach for asymmetric learning.

## 1 INTRODUCTION

Learning a task-relevant metric among samples is a common application of machine learning, with use in retrieval, clustering, and ranking. A classic example of retrieval is in visual recognition where, given an object image, the system tries to identify the class based on an existing labeled dataset. To do this, the model can learn a measure of similarity between pairs of images, assigning small distances between images of the same object type. Given the broad successes of deep learning, there has been a recent surge of interest in deep metric learning—using neural networks to automatically learn these similarities (Hoffer & Ailon, 2015; Huang et al., 2016; Zhang et al., 2020).

The traditional approach to deep metric learning learns an embedding function over the input space so that a simple distance measure between pairs of embeddings corresponds to task-relevant spatial relations between the inputs. The embedding function $f$ is computed by a neural network, which is learned to encode those spatial relations. For example, we can use the basic Euclidean distance metric to measure the distance between two samples $x$ and $y$ as $\|f(x) - f(y)\|_2$. This distance is critical in two ways. First, it is used to define the loss functions, such as triplet or contrastive loss, to dictate *how* this distance should be used to capture task-relevant properties of the input space. Second, since $f$ is trained to optimize the loss function, the distance influences the learned embedding $f$.

This approach has limitations. When the underlying reference distance is asymmetric or does not follow the triangle inequality, a standard metric cannot accurately capture the data. An important example is clustering over probability distributions, where the standard k-means approach with Euclidean distance is sub-optimal, leading to alternatives being used like the KL-divergence (Banerjee et al., 2005). Other cases include textual entailment and learning graph distances which disobey the triangle inequality.

Recent work has shown interest in learning an appropriate distance from the data instead of pre-determining the final metric between embeddings (Cilingir et al., 2020; Pitis et al., 2020). A natural class of distances that include common measures such as the squared Euclidean distance are the Bregman divergences (Bregman, 1967). They are parametrized by a strictly convex function $\phi$ and measure the distance between two points $x$ and $y$ as the first-order Taylor approximation error of the

| Method | $\phi$ representation | Learning Approach | Complexity | Joint Learning |
|--------|---------------------|-------------------|------------|----------------|
| NBD | $\phi(x) = \text{ICNN}(x)$ | Gradient Descent | $\mathcal{O}(|\theta|)$ | Yes |
| PBDL | $\phi(x) = \max_i(b_i^\top x + z_i)$ | Linear Programming | $\mathcal{O}(n^3)$ | No |
| Deep-div | $\phi(x) = \max_i(b_i^\top x + z_i)$ | Gradient Descent | $\mathcal{O}(|\theta| + K)$ | Yes |

Table 1: Comparison of our Bregman learning approach NBD with prior methods. NBD simultaneously has better representational power and computational efficiency.

function originating from $y$ at $x$. The current best approach in Bregman learning approximates $\phi$ using the maximum of affine hyperplanes (Siahkamari et al., 2020; Cilingir et al., 2020).

In this work we address significant limitations of previous approaches and present our solution, Neural Bregman Divergences (NBD). NBD is the first non max-affine approach to learn a deep Bregman divergence, avoiding key limitations of prior works. We instead directly model the generating function $\phi(x)$, and then use $\phi(x)$ to implement the full divergence $D_\phi$. To demonstrate efficacy, we leverage prior and propose several new benchmarks of asymmetry organized into three types of information they provide: 1) quality of learning a Bregman divergence directly, 2) ability to learn a Bregman divergence and a feature extractor jointly, and 3) effectiveness in asymmetric tasks where the ground truth is known to be non-Bregman. This set of tests shows how NBD is far more efficacious in representing actual Bregman divergences than prior works, while simultaneously performing better in non-Bregman learning tasks.

The rest of our paper is organized as follows. In §2 we show how to implement NBD using an Input Convex Neural Network and compare to related work, with further related work in §3. In §4 we demonstrate experiments where the underlying goal is to learn a known Bregman measure, and then to jointly learn a Bregman measure with an embedding of the data from which the measure is computed. Then §5 studies the performance of our method on asymmetric tasks where the underlying metric is not Bregman, to show more general utility where prior Bregman methods fail. Finally we conclude in §6.

## 2 NEURAL BREGMAN DIVERGENCE LEARNING

A Bregman divergence computes the divergence between two points $x$ and $y$ from a space $\mathcal{X}$ by taking the first-order Taylor approximation of a generating function $\phi$. This generating function is defined over $\mathcal{X}$ and can be thought of as (re-)encoding points from $\mathcal{X}$. A proper and informative $\phi$ is incredibly important: different $\phi$ can capture different properties of the spaces over which they are defined. Our aim in this paper is to learn Bregman divergences by providing a neural method for learning informative functions $\phi$.

**Definition 2.1.** *Let $x, y \in \mathcal{X}$, where $\mathcal{X} \subseteq \mathbb{R}^d$. Given a continuously differentiable, strictly convex $\phi : \mathcal{X} \to \mathbb{R}$, the Bregman divergence parametrized by $\phi$ is*

$$D_\phi(x, y) = \phi(x) - \phi(y) - \langle \nabla\phi(y), x - y \rangle, \tag{1}$$

*where $\langle \cdot, \cdot \rangle$ represents the dot product and $\nabla\phi(y)$ is the gradient of $\phi$ evaluated at $y$.*

A properly defined $\phi$ can capture critical, inherent properties of the underlying space. By learning $\phi$ via Eq. (1), we aim to automatically learn these properties. For example, Bregman divergences can capture asymmetrical relations: if $\mathcal{X}$ is the $D$-dimensional simplex representing $D$-dimensional discrete probability distributions then $\phi(x) = \langle x, \log x \rangle$ yields the KL divergence, $D_\phi(x, y) = \sum_d x_d \log \frac{x_d}{y_d}$. On the other hand, if $\mathcal{X} = \mathbb{R}^d$ and $\phi$ is the squared $L_2$ norm ($\phi(y) = \|y\|_2^2$), then $D_\phi(x, y) = \|x - y\|_2^2$. Focusing on the hypothesis space of Bregman divergences is valuable due to the fact that many core machine learning measures, including squared Euclidean, Kullback-Leibler, and Ikura-Saito divergences, are special cases of Bregman divergences. While special cases of the Bregman divergence are used today, and many general results have been proven over the space of Bregman measures, less progress has been made in *learning* Bregman divergences.

### 2.1 EXISTING BREGMAN LEARNING APPROACHES

**PBDL.** Recent works have proposed ways to empirically learn the Bregman divergence that best represents a dataset by focusing on a max-affine representation of $\phi$ (Siahkamari et al., 2020;

2022). Given a set of $K$ affine hyperplanes of the form $b_i^\top x + z_i$, the convex function $\phi$ can be given a lower-bound approximation as $\phi(x) = \max_i (b_i^\top x + z_i)$. Given a dataset of $m$ distance constraints between pairs of samples $D(x_a, x_p) \leq D(x_a, x_n)$, the parameters $\{b_i, z_i\}_1^K$ can be learned using convex optimization to minimize an objective function, e.g. a variant of the triplet loss $\mathcal{L}_{tr}(x_a, x_p, x_n) = \sum_{j=1}^m \max\{0, 1 + D(x_{a_j}, x_{p_j}) - D(x_{a_j}, x_{n_j})\}$. With some rewriting of the objective, the problem can be solved with techniques such as ADMM. We have modified some of the presentation for simplicity; refer to the original works for detail.

While effective on smaller datasets at learning a suitable divergence for ranking and clustering, this approach has limitations when scaling up to the dataset sizes used in deep learning. It requires a fixed set of triplets for training which limits the data size (whereas on-demand mining can generate $\mathcal{O}(n^3)$ triplets). Even though ADMM can be distributed (and the 2-block ADMM from Siahkamari et al. (2022) is an order of magnitude faster than the original), it does not scale as well as batch gradient descent methods to large data. Moreover, it cannot be directly used for deep learning tasks, as we cannot simultaneously learn an embedding with the divergence.

**Deep-div.** Following the max-affine approach, Cilingir et al. (2020) proposed the first deep learning approach to learn $\phi$, where the base neural network is followed by a layer of $K$ max-affine components. Each max-affine component is a linear layer (or a shallow network), trained via backpropagation. Recall that $\phi(x) := \max_i (b_i^\top x + z_i)$ and let $\phi_i(x) = b_i^\top x + z_i$ for each $i$. To simplify the computation of $d_\phi(x, y)$, they make use of the following property.

**Fact 2.1.1.** *Let $i$ and $j$ be the corresponding max-affine components for $\phi(x)$ and $\phi(y)$. Then the Bregman divergence between two points $x$ and $y$ is $D_\phi(x, y) = \phi_i(x) - \phi_j(x)$.*

Thus for a given $x, y$, the components $i, j$ are updated by backpropagation. Observe that no matter how large $K$ is, $\phi$ is neither continuously differentiable nor strictly convex. As a result, this structure is not reliably learned using gradient descent. We provide an example to illustrate this, which is confirmed in our experiments (e.g. Table 4).

**Example 2.2.** *Consider a Bregman regression problem learning $D(x, y)$ from known targets $d$ and loss function $\mathcal{L}(x, y, d; \phi) = (D_\phi(x, y) - d)^2$. The gradient with respect to each parameter $b$ is*

$$\nabla_b \mathcal{L}(x, y, d; \phi) = 2(D_\phi(x, y) - d) \cdot \nabla_b D_\phi(x, y)$$

*For any max-affine slopes $b_i$, $b_j$, $\nabla_b D_\phi(x, y)$ are $x$ and $-x$ respectively, while the others are 0. If we are also learning an embedding $f(x)$, then replace $x$ with $f(x)$ above, and furthermore for any embedding parameter $\theta$, we have $\nabla_\theta D_\phi(x, y) = (b_i - b_j) \cdot \nabla_\theta f(x)$.*

*From this we infer that when $i = j$ no learning occurs. Furthermore, if any max-affine hyperplane $l$ is (nearly) dominated by the others over the domain of the inputs (that is, $\phi_l(x) < \phi(x)$ for (almost) all $x$), that component never (rarely) gets updated, resulting in essentially 'dead' components.*

Another implication is that $d_\phi(x, y) = 0$ for any $x, y$ located on the same maximal component. By linearity there are at most $K$ such regions, thus in classification or clustering with $K$ classes the run-time of Deep-div is forced to increase with $K$. Even so, this resolution is low for more fine-grained tasks such as regression unless $K$ is very large. In practice, training this method on deep metric learning or regression shows issues which corroborate our analysis.

## 2.2 Representing $D_\phi$ via $\phi$ directly

Next we describe our new method, where we directly learn $\phi$ with a continuous convex neural network, which has a number of advantages over the piecewise approach. Whereas each pass of Deep-div requires an $\mathcal{O}(K)$ loop over the max-affine components, slowing training and inference, our approach does not incur additional complexity beyond an additional backpropagation step. Our approach is suited for both classification and regression tasks because it gives much finer resolution to $\phi$, while max-affine approaches incur irreducible error when the target is smooth. This is further improved by selecting an activation function to make the learned $\phi$ strictly convex, continuously differentiable, unlike the max-affine approximation. Empirically, we demonstrate that our method converges consistently and more efficiently to lower error solutions.

To represent $\phi$, we adopt the Input Convex Neural Network (ICNN) (Amos et al., 2017). The ICNN composes linear layers with non-negative weights $W^+$ and affine functions with unconstrained weights $U$ with convex non-decreasing activation functions $g(\cdot)$. The composition of these three

components for the $i$th layer of an ICNN is given by Eq. (2), with $z_i$ the $i$'th layer's input and $z_{i+1}$ the output,

$$z_{i+1} = g\left(W_i^+ z_i + U_i z_0 + b_i\right). \tag{2}$$

By construction, the resulting neural network satisfies convexity. Chen et al. (2019) and Pitis et al. (2020) have shown under specific conditions that ICNNs universally approximate convex functions. Furthermore, with hidden layers, the representational power of an ICNN is exponentially greater than that of a max-affine function ( Chen et al. (2019) Thm. 2), giving much finer resolution over $\phi$.

Prior works on the ICNN have only tried piecewise linear activation functions such as the ReLU variants for $g(\cdot) = \max(x, 0)$; we instead use the Softplus activation $g(x) = \log(1 + \exp(x))$ which lends the network smoothness and strict convexity. This is an important design choice as learning $\nabla\phi(y)$ involves the second derivatives, which for any piecewise activation is zero. This causes vanishing gradients in the $\langle\nabla\phi(y), x - y\rangle$ term, restricting the capacity to learn. We further discuss its representational capacity in F.1.

**Efficient Computation.** In order to backpropagate a loss through the $\nabla\phi(y)$ term in 1, we use double backpropagation as in Drucker & Le Cun (1991). Normally computing the gradient of $\nabla\phi(y)$ would involve constructing the Hessian, with a resulting quadratic increase in computation and memory use. Double backpropagation uses automatic differentiation to efficiently compute gradients with respect to the inputs with the "Jacobian vector product" (Frostig et al., 2021), so that $\langle\nabla\phi(y), x - y\rangle$ can be computed in the cost of evaluating $\phi(y)$ one additional time. Since there are already three calls to $\phi$, this is only a 25% increase in computational overhead to backpropagate through Eq. (1), provided we have a learnable representation of $\phi$. This functionality has been implemented in the PyTorch API (Paszke et al., 2017). Furthermore, we can vectorize the Bregman divergence over a pairwise matrix, making our method *significantly more computationally efficient* than prior approaches. In particular, avoiding the $\mathcal{O}(K)$ max-affine computation of Deep-div cuts our training time in half (Table 8). We show empirical timings, along with discussion, in Appendix A.

## 2.3 JOINT TRAINING

The original feature space is rarely ideal for computing the distance measures between samples. Classical metric learning generally attempts to apply a linear transformation to the feature space in order to apply a fixed distance function $D(\cdot, \cdot)$ such as Euclidean distance (Jain et al., 2012; Kulis et al., 2012). In deep metric learning, a neural network $f_\theta$ is used to embed the samples into a latent space where the distance function is more useful (Musgrave et al., 2020). In our approach, instead of fixing the distance function, we also learn a Bregman divergence as the measure:

$$D_\phi(f_\theta(x), f_\theta(y)) = \phi(f_\theta(x)) - \phi(f_\theta(y)) - \langle\nabla\phi(\tilde{y}), f_\theta(x) - f_\theta(y)\rangle \tag{3}$$

with $\nabla\phi$ evaluated at $\tilde{y} = f_\theta(y)$.

Note we now have two sets of parameters to learn: those associated with $\phi$ and the encoder ($\theta$). During training, they are simultaneously learned through gradient descent, which involves double-backpropagation as described earlier. We summarize this process in Alg. 1. The metric model accepts two samples as input and estimates the divergence between them. When the target divergence value is available, the metric can be trained using a regression loss function such as mean square error. Otherwise, an implicit comparison such as triplet or contrastive loss can be used.

## 3 OTHER

## RELATED WORK

In classic metric learning methods, a linear or kernel transform on the ambient feature space is used, combined with a standard distance

---

**Algorithm 1 Neural Bregman Divergence (NBD)**. Given data pairs $(a_i, b_i)$, our approach learns (1) $f_\theta$ to featurize $a_i$ and $b_i$; (2) $\phi$ to compute a Bregman divergence value $\hat{y}$ between the featurized data points. The computed Bregman divergence is trained via loss function $\ell$ to be close to a *target* divergence value $y_i$. If a target divergence value isn't available, an implicit loss function can be used.

---

**Require:** Dataset of pairs and target distance, Loss function $\ell(\cdot, \cdot) : \mathbb{R} \to \mathbb{R}$
1: $f_\theta \leftarrow$ arbitrary neural network as a feature extractor
2: $\phi \leftarrow$ a ICNN network parameterized as by Eq. (2)
3: **for** each data tuple $(\boldsymbol{a}_i, \boldsymbol{b}_i)$ with label $y_i$ in dataset **do**
4:      $\boldsymbol{x} \leftarrow f_\theta(\boldsymbol{a}_i)$          ▷ Perform feature extraction
5:      $\boldsymbol{y} \leftarrow f_\theta(\boldsymbol{b}_i)$
6:      $rhs \leftarrow \langle\nabla\phi(\boldsymbol{y}), \boldsymbol{x} - \boldsymbol{y}\rangle$    ▷ Use double backprop
7:      $\hat{y} \leftarrow \phi(\boldsymbol{x}) - \phi(\boldsymbol{y}) - rhs$    ▷ Empirical Bregman
8:      $\ell(\hat{y}, y_i).\,\text{backward}()$       ▷ Compute gradients
9:      update parameters of $\phi$ and $\theta$
10: **return** Jointly trained feature extractor $f_\theta$ and learned Bregman Divergence $\phi$

---

| Model | Exponential | | Gaussian | | Multinomial | |
|---|---|---|---|---|---|---|
| | Purity | Rand Index | Purity | Rand Index | Purity | Rand Index |
| NBD | **0.735** $_{0.08}$ | **0.830** $_{0.03}$ | **0.913** $_{0.05}$ | **0.938** $_{0.03}$ | **0.921** $_{0.02}$ | **0.939** $_{0.01}$ |
| Deep-div | 0.665 $_{0.12}$ | 0.788 $_{0.08}$ | 0.867 $_{0.12}$ | 0.910 $_{0.07}$ | 0.876 $_{0.08}$ | 0.919 $_{0.04}$ |
| Euclidean | 0.365 $_{0.02}$ | 0.615 $_{0.02}$ | 0.782 $_{0.11}$ | 0.869 $_{0.05}$ | 0.846 $_{0.09}$ | 0.900 $_{0.05}$ |
| Mahalanobis | 0.452 $_{0.05}$ | 0.697 $_{0.02}$ | 0.908 $_{0.06}$ | 0.935 $_{0.03}$ | 0.894 $_{0.06}$ | 0.926 $_{0.03}$ |
| PBDL | 0.718 $_{0.08}$ | **0.830** $_{0.04}$ | 0.806 $_{0.14}$ | 0.874 $_{0.09}$ | 0.833 $_{0.08}$ | 0.895 $_{0.04}$ |

Table 2: We cluster data generated from a mixture of exponential, Gaussian, and multinomial distributions. Learning the metric from data is superior to using a standard metric such as Euclidean. Our approach NBD furthermore outperforms all other divergence learning methods. Means and standard deviations are reported over 10 runs.

function such as Euclidean or cosine distance. The linear case is equivalent to Mahalanobis distance learning. Information on such approaches are in (Xing et al., 2002; Kulis et al., 2012; Jain et al., 2012; Kulis et al., 2009). Bregman divergences generalize many standard distance measures and can further introduce useful properties such as asymmetry. They have classically been used in machine learning for clustering, by modifying the distance metric used in common algorithms such as kmeans (Banerjee et al., 2005; Wu et al., 2009). One of the first methods to learn a Bregman divergence fits a non-parametric kernel to give a local Mahalanobis metric. The coefficients for the data points are fitted using subgradient descent (Wu et al., 2009). We described the more recent approaches earlier.

Recently Pitis et al. (2020) approached asymmetric distance learning by fitting a norm $N$ with modified neural networks which satisfy norm properties and using the induced distance metric $N(x - y)$. They introduce two versions that we include as baselines: one (*Deepnorm*) parametrizes $N$ with a modified ICNN that satisfies properties such as non-negativity and subadditivity. The second (*Widenorm*) computes a nonlinear transformation of a set of Mahalanobis norms. By construction, these metrics allow for asymmetry but still satisfy the triangle inequality. On the other hand, the Bregman divergence does not necessarily obey the triangle inequality. This is appealing for situations where the triangle inequality may be too restrictive.

## 4 BREGMAN EXPERIMENTS

We conduct several experiments that validate our approach as an effective means of learning divergences across a number of tasks. *Over 44 comparisons, NBD outperforms prior Bregman learning approaches in all but three.* In the first section §4.1 we demonstrate that NBD *effectively learns standard Bregman retrieval and clustering benchmarks*, outperforming the previous Bregman methods PBDL and Deep-div. In addition, we construct a Bregman regression task in §4.2 where the labels are known divergences over raw feature vectors, so that the *only learning task is that of the divergence itself*. Finally in §4.3 we investigate the ability of our method to *learn the ground truth divergence while simultaneously learning to extract a needed representation*, training a sub-network's parameters $\theta$ and our divergence $\phi$ jointly. This is typified by the "BregMNIST" benchmark, which combines learning the MNIST digits with the only supervisory signal being the ground truth divergence between the digit values. Refer to the Appendix for detailed training protocols and data generation procedures.

| Dataset | Model | MAP | AUC | Purity | Rand |
|---|---|---|---|---|---|
| abalone | Deep-div | 0.281 | 0.645 | 0.377 | 0.660 |
| | Euclidean | 0.301 | 0.666 | 0.422 | **0.750** |
| | Mahalanobis | 0.310 | 0.677 | 0.419 | **0.750** |
| | NBD | **0.316** | **0.682** | **0.432** | **0.750** |
| | PBDL | 0.307 | 0.659 | 0.386 | 0.735 |
| balance scale | Deep-div | 0.804 | 0.859 | 0.869 | 0.828 |
| | Euclidean | 0.611 | 0.666 | 0.633 | 0.568 |
| | Mahalanobis | 0.822 | 0.854 | 0.851 | 0.761 |
| | NBD | **0.887** | **0.915** | **0.898** | **0.872** |
| | PBDL | 0.836 | 0.855 | 0.872 | 0.814 |
| car | Deep-div | 0.787 | 0.757 | 0.852 | 0.750 |
| | Euclidean | 0.681 | 0.589 | 0.704 | 0.523 |
| | Mahalanobis | 0.787 | 0.752 | 0.778 | 0.654 |
| | NBD | **0.820** | **0.803** | **0.860** | **0.758** |
| | PBDL | 0.798 | 0.775 | 0.854 | 0.750 |
| iris | Deep-div | 0.945 | 0.967 | 0.811 | 0.820 |
| | Euclidean | 0.827 | 0.897 | 0.820 | 0.828 |
| | Mahalanobis | 0.946 | 0.973 | 0.884 | 0.879 |
| | NBD | **0.957** | **0.977** | **0.909** | **0.902** |
| | PBDL | 0.943 | 0.967 | 0.889 | 0.888 |

Table 3: On real datasets, learned Bregman divergences outperform Euclidean or Mahalanobis metrics for downstream ranking (MAP, AUC) and clustering (Purity, Rand Index). NBD beats prior Bregman learning approaches on most datasets. See Appendix Table 10 for standard deviations and more datasets.

## 4.1 BREGMAN RANKING AND CLUSTERING

Our first task expands the distributional clustering experiments in Banerjee et al. (2005); Cilingir et al. (2020). The datasets consist of mixtures of $N = 1000$ points in $\mathbb{R}^{10}$ from five clusters, where the multivariate distribution given cluster identity is non-isotropic Gaussian, exponential, or multinomial. Given a distance metric, we apply a generalized k-means algorithm to cluster the data points. While standard metrics, such as L2-distance and KL-divergence, may be ideal for specific forms of data (e.g. isotropic Gaussian and simplex data, respectively), our goal is to learn an appropriate metric directly from a separate labeled training set. In particular, because Bregman divergences are uniquely associated with each member of the exponential family (Banerjee et al., 2005), our method is especially suited for clustering data from a wide range of distributions which may not be known ahead of time. To learn the metric from data, we apply triplet mining, including all triplets with non-zero loss (Hoffer & Ailon, 2015). We use the same method to train all models except for the Euclidean baseline, which requires no training, and PBDL where we directly use the authors' code.

As shown in Table 2, our method NBD gives improved clustering over all distributions compared to all baselines. In particular, standard k-means with Euclidean distance is clearly inadequate. While the Mahalanobis baseline shows significant improvement, it is only comparable to NBD in the Gaussian case, where a matrix can be learned to scale the clusters to be isotropic. This task indicates the importance of learning flexible divergences from data.

After demonstrating success in distributional clustering, we now apply our method to ranking and clustering real data (Table 3), as first shown in Siahkamari et al. (2020). For the ranking tasks, the test set is treated as queries for which the learned model retrieves items from the training set in order of increasing divergence. The ranking is scored using mean average precision (MAP) and area under ROC curve (AUC). Our method again outperforms the other Bregman learning methods in the large majority of datasets and metrics. We emphasize that these are standard experiments from recent work, on which our method proves superior.

## 4.2 DIVERGENCE REGRESSION

| Truth | Euclidean | | | Mahalanobis | | | $x \log x$ | | | KL | | |
|---|---|---|---|---|---|---|---|---|---|---|---|---|
| Correlation | None | Med | High | None | Med | High | None | Med | High | None | Med | High |
| NBD | *0.17* | *0.15* | *0.16* | *0.16* | *0.18* | *0.20* | **0.52** | **0.54** | **0.57** | **0.19** | **0.19** | **0.19** |
| Deep-div | 7.78 | 7.81 | 7.84 | 17.92 | 12.26 | 14.15 | 2.59 | 2.67 | 2.70 | 0.44 | 0.50 | 0.51 |
| Deepnorm | 3.56 | 3.97 | 4.15 | 7.70 | 5.97 | 7.66 | 1.59 | 1.74 | 1.79 | 0.30 | 0.28 | 0.28 |
| Widenorm | 3.56 | 3.99 | 4.12 | 7.73 | 6.01 | 7.60 | 1.49 | *1.48* | *1.48* | 0.30 | 0.28 | 0.28 |
| Mahalanobis | **0.00** | **0.03** | **0.05** | **0.02** | **0.04** | **0.09** | *1.45* | 1.67 | 1.72 | *0.23* | *0.22* | *0.22* |

Table 4: Regression test MAE when unused distractor features are correlated (None/Med/High) with the true/used features. Best results in **bold**, second best in *italics*. NBD performs best on asymmetric regression, and second-best to Mahalanobis on symmetric regression, where a Mahalanobis distance is expected to fit perfectly.

As a confirmation that our method can faithfully represent Bregman divergences, we use simulated data to demonstrate that our method efficiently learns divergences between pairs of inputs. We generate pairs of 20-dim. vectors from a Normal distribution, with 10 informative features used to compute the target divergence and 10 distractors. To be more challenging and realistic, we add various levels of correlations among all features to make the informative features harder to separate.

The following target divergences are used: (1) squared Euclidean distance (symmetric); (2) squared Mahalanobis distance (symmetric); (3) $\phi(x) = x \log x$ (asymmetric); (4) KL-divergence (asymmetric). In this task we compare our NBD with Deep-div and Mahalanobis, but we did not find a regression option for PBDL in the authors' code. Instead we add Deepnorm and Widenorm metrics from Pitis et al. (2020) as alternative baselines which do not learn Bregman divergences.

The results of these experiments are in Table 4, with loss curves shown in Appendix Fig. 4. In the symmetric cases of the Euclidean and Mahalanobis ground-truth, our NBD method performs nearly as well as using a Mahalanobis distance itself. This shows that our method is not losing any representational capacity in being able to represent these standard measures. This is notably not true for the prior approaches for asymmetric learning: Deepnorm, Widenorm, and Deep-div.

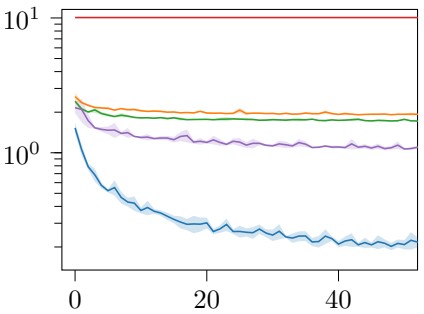
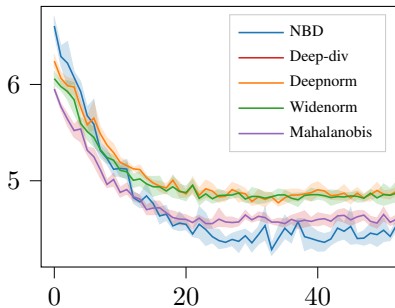

Figure 2: MSE (y-axis) after epochs of training (x-axis) on asymmetric BregMNIST (left) and BregCIFAR (right) with true $\phi(x) = x \log x$. NBD performs best in both tasks.

Notably, unlike in the clustering tasks, the piecewise representation of $\phi$ in Deep-div is unable to accurately represent Bregman regression targets, as discussed earlier in §2.1. In Fig. 4c and Fig. 4d two asymmetric divergences are used, and our NBD approach performs better than all existing options. Because these experiments isolate purely the issue of learning the divergence itself, we have strong evidence that our approach is the most faithful to learning a known divergence from a supervisory signal. Note that the Mahalanobis distance performed second best under all noise levels, meaning the prior asymmetric methods were in fact less accurate at learning asymmetric measures than a purely symmetric model.

### 4.3 CO-LEARNING AN EMBEDDING WITH A DIVERGENCE

Having shown that our method outperforms the prior Bregman learning approaches on shallow clustering, classification, and regression, we introduce a more challenging task, BregMNIST, where a neural embedding must be learned along with the divergence metric. The dataset consists of paired MNIST images, with the target distance being a Bregman divergence between the digits shown in the images. Example pairs are displayed in Fig. 1 for the asymmetrical Bregman divergence parametrized by $\phi(x) = (x + 1) \log(x + 1)$.

We also make a harder version by substituting MNIST with CIFAR10 with the same divergence labels. In both cases the relation of features to class label is arbitrary (that is, we impose an ordinal relation among labels that does not exist in the data), meaning that the embedding function must learn to effectively map image classes to the correct number used to compute the divergence, while the metric head must also learn to compute the target Bregman divergence. The results of the experiments (Fig. 2) mirror our results in §4.2. For both BregMNIST and BregCIFAR NBD performs best, while prior methods of learning asymmetric measures perform worse than the Mahalanobis distance.

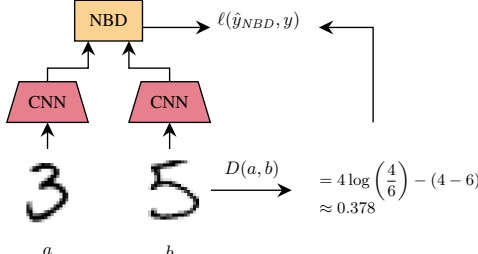

Figure 1: Demonstration of the BregMNIST task. Nodes with the same color indicate weight sharing. Each image is embedded by a CNN, and the ground-truth divergence is computed from the digit values of the input images. The embeddings of each image are given to NBD, and the loss is computed from NBD's output and the true divergence. The CNN and NBD are learned jointly.

## 5 NON-BREGMAN LEARNING

We have shown that our NBD method is the most effective among all available options when the underlying ground-truth is from the class of Bregman divergences. In this section we will now explore the effectiveness of our approach on tasks that are known to be non-Euclidean, but not necessarily representable by a Bregman divergence. The purpose of these experiments is to show that NBD does not depend on the underlying representation being a proper divergence in order to still be reasonably effective, and that it is still more effective then the prior Deep-div approach to Bregman learning. This is also of practical relevance to applications: just as the Euclidean metric was used for convenient properties and simplicity, without belief that the underlying system was truly Euclidean, our NBD

may be valuable for developing more flexible methods that inherit the mathematical convenience of Bregman divergences. These tasks probe the efficacy of the closest Bregman approximation of the underlying divergence, so we expect that our method will not surpass SOTA when the task is sufficiently non-Bregman.

## 5.1 DEEP METRIC LEARNING

We extend the ranking experiments from §4.1 and the architecture of Fig. 1 to the deep metric learning setting where an embedding is learned alongside the divergence. We use a ResNet-18 as the base feature extractor and apply batch triplet mining to learn Eq. 3 by minimizing the triplet loss.

|          | CIFAR10 | STL10 | SVHN |
|----------|---------|-------|------|
| NBD      | **95.6** | 95.0  | **96.9** |
| Deep-div | 93.3    | 92.3  | 96.0 |
| Deepnorm | 95.0    | 95.0  | **96.9** |
| Widenorm | **95.6** | **95.1** | **96.9** |
| Euclidean | 95.0   | 95.0  | **96.9** |

Table 5: MAP@10 on deep metric learning, replacing the standard Euclidean distance with a metric co-learned with the embedding.

Most metrics perform comparably in this experiment, although Deep-div is consistently outperformed by the others. We observed that Deep-div has higher variance, and depending on the initialization could learn well or not at all. This may be due to 'dead' affine components discussed earlier. Fixing the Euclidean distance still appears as effective as learning the final metric here. We hypothesize this is because the embedding space of image datasets is well-behaved enough for a standard distance to accurately cluster images. In the following experiments we will investigate tasks where, even after applying an arbitrary feature extractor, a standard distance measure is no longer sufficient.

## 5.2 APPROXIMATE SEMANTIC DISTANCE

The next task involves learning symmetric distances that do not follow the triangle inequality. We group the CIFAR10 classes into two categories: man-made and natural. Within each category we select an arbitrary exemplar class (*car* and *deer* in our experiment). We then assign proxy distances between classes to reflect semantic similarity: 0.5 within the same class, 2 between any non-exemplar class and its category exemplar, and 8 between non-exemplar classes within a category. Pairs from different categories are not compared. Besides disobeying the triangle inequality, the distance values do not reflect a known divergence and can be changed.

Like BregCIFAR, we present pairs of images to the model, which simultaneously adjusts a neural embedding and learn a divergence function such that inter-class distances in the embedding space match the target values. This task is harder than the previous ones because it is not sufficient to learn a separable embedding for each class; the embeddings must additionally be arranged appropriately in a non-Euclidean space. The results in Table 6 indicate our method effectively learns distances that do not follow the triangle inequality. The Deep-div approach does second-best here due to the small space of valid outputs. The other approaches by limitation adhere to the triangle inequality and do not perform as well.

| Metric      | Same   | Unseen |
|-------------|--------|--------|
| NBD         | **0.04** | **3.52** |
| Deep-div    | *0.10* | *4.13* |
| Deepnorm    | 1.23   | 4.18   |
| Widenorm    | 1.39   | 4.50   |
| Mahalanobis | 2.00   | 4.56   |

Table 6: MSE for CIFAR10 category semantic distance after 200 epochs. Our NBD performs the best on seen and unseen images.

## 5.3 OVERLAP DISTANCE

The overlap distance task presents pairs of the same image or different images, but with different crops taken out. A horizontal and vertical cut are chosen uniformly at random from each image. When the crops are based on the same image, the asymmetrical divergence measure between images $X$ and $Y$ is the percent intersection area: $D(X, Y) = 1 - \frac{|X \cap Y|}{|X|}$. Otherwise the divergence is 1. We use the INRIA Holidays dataset (see Appendix G). The results can be found in Fig. 3, where we see NBD performs the second best of all options.

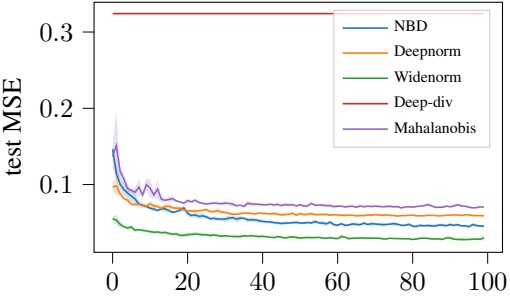

Figure 3: MSE (y-axis) for predicting the overlap $D(X, Y)$ between two image embeddings learned jointly with the underlying CNN.

| Method | 3d Train | 3d Test | 3dd Train | 3dd Test | octagon Train | octagon Test | taxi Train | taxi Test | traffic Train | traffic Test |
|---|---|---|---|---|---|---|---|---|---|---|
| NBD | 4.34 | 33.49 | 19.91 | 337.59 | 4.67 | 25.32 | 3.11 | 66.27 | 2.53 | 12.03 |
| Deepnorm | 4.97 | 22.44 | 34.40 | 275.99 | 4.81 | 15.19 | 1.52 | 20.31 | 1.76 | 5.27 |
| Mahalanobis | 4.45 | 30.90 | 24.99 | 267.18 | 6.82 | 44.30 | 1.31 | 18.32 | 1.47 | 5.60 |
| Deep-div | 695.97 | 930.57 | 589.13 | 806.14 | 879.94 | 1046.08 | 489.80 | 625.16 | 399.14 | 618.94 |
| Widenorm | 4.49 | 27.92 | 25.76 | 253.65 | 5.17 | 23.46 | 1.18 | 16.20 | 1.44 | 5.21 |
| Bregman-sqrt | 5.94 | 27.59 | 27.70 | 266.25 | 8.62 | 40.18 | 1.57 | 19.02 | 1.63 | 5.23 |
| Bregman-GS | 4.50 | 30.51 | 23.78 | 266.91 | 7.26 | 43.13 | 1.17 | 16.71 | 1.55 | 5.49 |

Table 7: Results of estimating distances on the shortest-path task. Triangle-inequality preserving deep and wide-norm are expected to perform best. Our NBD performs significantly better than previous Bregman learning approach Deep-div, and can be competitive with the triangle-inequality preserving methods. The gap between train and test loss shows the impact of triangle inequality helping to avoid over-fitting the observed sub-graph used for training. Dataset and experiment details in Appendix.

We observe that Widenorm performs better on this task, especially during the initial learning process, due to the fact that it is permitted to violate the positive definiteness property of norms: $D(x, x) > 0$. Thus the method learns to map a difference of zero between embeddings to some intermediate distance with lower MSE. This can be problematic in use cases where the definiteness is important.

### 5.4 Shortest path length

Our final task involves estimating the shortest path on a graph from one embedded node to another based on their distances to and from a set of landmark nodes. This task inherently favors the Widenorm and Deepnorm methods because they maintain the triangle inequality (i.e., no shortcuts allowed in shortest path), and so are expected to perform better than NBD. We reproduce the experimental setup of Pitis et al. (2020) closely with details in Appendix E.

The results for each method are shown in §5.4, which largely match our expectations. The triangle-inequality preserving measures usually perform best, given the nature of the problem: any violation of the triangle inequality means the distance measure is "taking a shortcut" through the graph search space, and thus must be under-estimating the true distance to the target node. NBD and Deep-div, by being restricted to the space of Bregman divergences, have no constraint that prevents violating the triangle-inequality, and thus often under-estimate the true distances. Comparing the train and test losses help to further assess this behavior, as the training pairs can be overfit to an extent. We see that NBD effectively learns the asymmetric relationships between seen points despite underestimating the distance to new points.

To further explore the degree to which the properties underlying Bregman divergences affect shortest path length learning, we introduce two extensions to NBD. The first is a soft modification encouraging the triangle inequality to be obeyed (Bregman-sqrt). The second has a hard constraint guaranteeing the triangle inequality (Bregman-GS) and is defined as $D_\phi^{gsb}(x, y) = D_\phi(x, y) + D_\phi(y, x) + \frac{1}{2}\|x - y\|_2^2 + \frac{1}{2}\|\nabla\phi(x) - \nabla\phi(y)\|_2^2$ (Acharyya et al., 2013). Results are included in §5.4, with detail on the extensions in the Appendix. There is an inherent tradeoff between the two extensions as Bregman-sqrt can be asymmetric but still does not require satisfying the triangle inequality, while Bregman-GS is symmetric but always satisfies the triangle inequality. We see that these two modifications to NBD are highly competitive with Deepnorm and Widenorm. Furthermore, the relative performance of each provides an indication of whether asymmetry or triangle inequality is more crucial to modeling a given dataset. These methods highlight that even when a given task is highly non-Bregman, NBD can be readily extended to relax or strengthen various assumptions to better model the data.

## 6 Conclusion

To enable future asymmetric modeling research, we have developed the Neural Bregman Divergence (NBD). NBD jointly learns a Bregman measure and a feature extracting neural network. We show that NBD learns divergences directly or indirectly when trained jointly with a network, and that NBD still learns effectively when the underlying metric is not a divergence, allowing effective use of our tool across a wide spectrum but retaining the nice properties of Bregman divergences.

## ACKNOWLEDGMENTS

We would like to thank the anonymous reviewers for their comments, questions, and suggestions. This material is based in part upon work supported by the National Science Foundation under Grant No. IIS-2024878, with some computation provided by the UMBC HPCF, supported by the National Science Foundation under Grant No. CNS-1920079. This material is also based on research that is in part supported by the Army Research Laboratory, Grant No. W911NF2120076, and by the Air Force Research Laboratory (AFRL), DARPA, for the KAIROS program under agreement number FA8750-19-2-1003. The U.S. Government is authorized to reproduce and distribute reprints for Governmental purposes notwithstanding any copyright notation thereon. The views and conclusions contained herein are those of the authors and should not be interpreted as necessarily representing the official policies or endorsements, either express or implied, of the Air Force Research Laboratory (AFRL), DARPA, or the U.S. Government.

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

## A   COMPUTATIONAL DISCUSSION

We measure timing information of NBD as well as the benchmarks used in our experiments for a divergence learning task as in §4.2. The training time (forward and backward passes), inference time (forward pass), and pairwise distance matrix computation are collected. While the pairwise distance is not directly used in our experiments, it is commonly computed in metric learning applications such as clustering and classification. The data dimension size $N \times D$ used here is characteristic of the size of many metric learning experiments, where the $N \times N$ pairwise distance matrix can be stored on GPU memory, but the $N \times N \times D$ tensor with embedding dimension $D$ does not fit: the fast but naive approach of flattening the matrix and passing as a $N \times N$-length batch does not work.

The Mahalanobis method is naturally the fastest method and serves as a reasonable runtime lower bound. This is because the distance can be expressed a simple composition of a norm with a linear layer. The squared Euclidean norm can be simplified as $D(x, y) = \|x\|_2^2 + \|y\|_2^2 - 2\langle x, y \rangle$, which can be efficiently computed. We can similarly compute the Bregman divergences. For example, the squared Euclidean distance is equivalently written as $D(x, y) = \|x\|_2^2 - \|y\|_2^2 - \langle 2y, x - y \rangle$, which is its Bregman divergence formulation. Though not necessary for our current experiments, the pairwise distances for larger batch sizes for the Mahalanobis and NBD can be readily implemented in PyKeOps Charlier et al. (2021). Thus the longer computational time for NBD can likely be attributed to the increased cost of double backpropagation and the convex metric architecture.

| Method | Training | Inference | Pairwise distance |
|---|---|---|---|
| NBD | 0.73 (0.08) | 0.10 (0.03) | 0.52 (0.06) |
| Deep-div | 1.63 (0.11) | 0.12 (0.02) | 0.59 (0.03) |
| Deepnorm | 0.81 (0.07) | 0.09 (0.02) | 4.53 (0.03) |
| Widenorm | 0.52 (0.04) | 0.08 (0.02) | 2.61 (0.03) |
| Mahalanobis | 0.46 (0.03) | 0.08 (0.02) | 0.40 (0.03) |

Table 8: Timing information for a divergence learning task as in §4.2, with embedding dimension 20 and a batch size of 1000, comparing the methods used in our experiments. We compute the per-epoch training time (forward and backward passes), inference time (forward pass), and pairwise distance matrix computation. Results are averaged over 30 runs, with standard deviation in parentheses.

On the other hand, the Deepnorm cannot be vectorized in such manner, so pairwise distances need to be computed on the order of $O(N^2)$ runtime, for example by further splitting the tensor into smaller mini-batches. While the Widenorm is composed of Mahalanobis metrics (set to 32 in our experiments) that can be vectorized, in our experiments the memory requirement was still too high, also requiring looping over sub-batches. We use sub-batch size 200 in this analysis. We note that the loop can be alternatively performed over the Mahalanobis components in Widenorm, but this would still be slower than the standard Mahalanobis and NBD methods. Finally, the Deep-div method efficiently computes pairwise distances, but its forward pass requires the input to be passed through a set of $K$ affine sub-networks via looping, increasing the computational time.

# B FIGURES FOR BREGMAN REGRESSION TASK

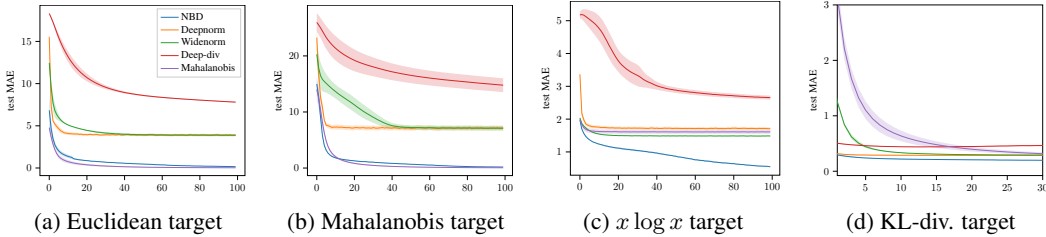

(a) Euclidean target      (b) Mahalanobis target      (c) $x \log x$ target      (d) KL-div. target

Figure 4: Results when vectors with 10 real features and 10 distractor features are used to compute different specific Bregman divergences. Mean absolute error is on the y-axis and number of training epochs on the x-axis. The shaded region around each plot shows the standard deviation of the results. Note in all cases our NBD has very low variance while effectively learning the target divergence.

## C    DATA GENERATION DETAILS

**Distributional clustering.** We sample 1000 points uniformly into 5 clusters, each with 10 feature dimensions. To generate non-isotropic Gaussians we sample means uniformly in the hyper-box within $[-4, 4]$ for each coordinate. The variances are a random PSD matrix (scikit-learn make_spd_matrix) added with $\sigma^2 I$ where $\sigma^2 = 5$. The reason for these values are because we aimed to have each mixture task be similarly difficult (clusters not perfectly separable but also not too challenging). For the multinomial task, we sampled 100 counts into the 10 feature dimensions. Each cluster's underlying probability distribution was sampled from $Dirichlet([10, 10, \ldots, 10])$. Finally, the exponential case are iid samples for each feature. The underlying cluster rates are sampled uniformly between [0.1, 10].

**Regression noise features**. To add correlation among features (both informative and distractors), we generate covariance matrices with controlled condition number $\kappa$ while keeping the marginal distributions of each feature as $x_i \sim \mathcal{N}(0, 1)$. For the medium correlation task $\kappa < 100$, while for high correlation $\kappa$ is between 250 and 500.

50000 pairs were generated with 20 features.

## D    EXPERIMENTAL PROCEDURE

**Overview.** Our experiments fall into two categories: regression (e.g. sections 4.2, 4.3, 5.2, 5.3, 5.4) and clustering/ranking (e.g. sections 4.1, 5.1). For the first category a scalar divergence target is assumed. Square error loss is used, and an input pair of samples is drawn from an underlying dataset or randomly generated. From the input pair, the true divergence target is computed. For example if the dataset has $N$ samples, there are $\mathcal{O}(N^2)$ possible pairs. Pairs are drawn simultaneously as batches with specified batch size, and an epoch (unless otherwise stated) is defined as the number of pairs equal to the size of the original dataset (so an epoch is the usual length, but not all data has necessarily been seen).

For the second category, a divergence target does not exist, only the relative ranking of anchor-positive being less distance than anchor-negative. The triplet loss is used here. In all such metric learning tasks, we fit models using triplet mining, with margin 0.2, and Adam optimizer. All triplets with non-zero loss are mined from a batch. For more detail on triplet loss and mining see Musgrave et al. (2020). Thus for a batch size of $B$ there are up to $\mathcal{O}(B^3)$ triplets. We iterate over the dataset in batches. The exception is PBDL which uses the authors' original code with pre-generated tuples.

**Distributional clustering.** We used batch size 128, 200 epochs, 1e-3 learning rate for all models. Here, and in all subsequent experiments, to train PBDL we used the authors' provided Python code, which uses the alternating direction method of multipliers (ADMM) technique. (They also provide Matlab code using Gurobi.)

**Bregman ranking.** Since Deep-div and NBD are deep learning approaches, we use Adam to optimize this problem instead of convex optimization solvers. To ensure convergence, we tune the learning rate and number of epochs using gridsearch over a validation set separated from the training data. We do the same for the Mahalanobis approach. A typical example of the parameters is batch size 256, 250 epochs, learning rate 1e-3.

**Regression.** We used 100 epochs of training with learning rate 1e-3, batch size 1000.

**Deep regression experiments.** For the remaining experiments which involve co-learning an embedding, we use default hyperparameter settings to keep methods comparable, such as Adam optimizer, learning rate 1e-3, batch size 128, embedding dimension 128, and 200 epochs. By deep regression, we refer to tasks that have a continuous target, such as BregMNIST, overlap distance, and shortest path.

For the MNIST/CIFAR tasks the embedding network consists of two/four convolutional layers respectively followed by two fully-connected layers (more specific details follow).For the semantic distance CIFAR task, we used a pretrained ResNet20 as the embedding without freezing any layers for faster learning He et al. (2016). Results were robust to the embedding model chosen.

We replicated each training and reported means and standard deviations. For the Bregman benchmark tasks we trained 20x, while for the deep learning/graph learning tasks we trained 5x. Learning curves

| Dataset | Asymmetric | Dimension | Edge weights | Details |
|---------|-----------|-----------|--------------|---------|
| 3d | No | 50x50x50 cubic grid | uniform $\{0.01, 0.02, \ldots, 1.00\}$ | edges wrap around |
| taxi | No | 25x25x25x25 (two objects on 2d grid) | uniform $\{0.01, 0.02, \ldots, 1.00\}$ | no wrap |
| 3dd | Yes | 50x50x50 cubic grid | uniform $\{0.01, 0.02, \ldots, 1.00\}$ | only one edge in each dimension is available |
| traffic | Yes | 100x100 2d grid | forward and reverse sampled from Normal with mean from uniform $\{0.01, 0.02, \ldots, 1.00\}$ | no wrap |
| octagon | Yes | 100x100 2d grid, diagonals connected | forward and reverse sampled from Normal with mean from uniform $\{0.01, 0.02, \ldots, 1.00\}$ | no wrap |

Table 9: Details of the shortest-path datasets, the number of dimensions in the graph, and how the edge weights in the graph are computed. These tasks were originally proposed by Pitis et al. (2020) and favor asymmetric methods that maintain the triangle inequality.

in the figures show mean and 95% confidence interval for the loss over each epoch. We used Quadro RTX 6000 GPUs to train our models.

**Deep metric learning.** Finally, this refers to the triplet loss experiment with co-learning of embedding and metric. For this we use the following settings: learning rate 1e-4, Adam optimizer, batch size 64, embedding dimension 32, 30 epochs. For all datasets that are 32x32, we resized to 224x224 and used a pretrained ResNet18 as the base embedding model, which is then finetuned during training. We used smaller learning rate and embedding/batch dimensions in keeping with the standard metric learning protocol in Musgrave et al. (2020) which we found gave more stable results.

We note that in Cilingir et al. (2020) they ran their Deep-div on a similar task and reported some results better and some worse than our results with Deep-div. We note that there are differences in protocol, where they extensively tuned the training procedure with hyperparmater optimization, whereas we selected default values to ensure robustness. However, we used a larger base extractor with pretrained weights whereas they used a custom CNN. As a result differences in performance can be expected. While our reported metric is MAP@10 and theirs was nearest neighbor accuracy, we found both measures to be very close across our experiments.

# E  SHORTEST PATH DETAILS

Pitis et al. (2020) introduced learning shortest path length in graphs as a task. The problem consists of learning $d(x, y)$ where $x, y \in \mathcal{V}$ are a pair of nodes in a large weighted graph $G = (\mathcal{V}, \mathcal{E})$. For each node, the predictive features consist of shortest distances from the node to a set of 32 landmark nodes (and vice versa for asymmetric graphs). As this task requires predicting the distance from one node to another, maintaining the triangle inequality has inherent advantages and is the correct inductive bias for the task. We still find the task useful to elucidate the difference between NBD and the prior Deep-div.

We reproduce the experimental setup of Pitis et al. (2020) closely, collecting a 150K random subset of pairs from the graph as the dataset, with true distances computed using $A^*$ search. While the original work normalized distances to mean 50, we found that such large regression outputs were difficult to learn. Instead we normalize to mean 1, which results in faster convergence of all methods. A 50K/10K train-test split was used. The features are standardized with added noise sampled from $\mathcal{N}(0, 0.2)$, and 96 normal-distributed distractor features were included. For additional details refer to Appendix E of Pitis et al. (2020). However, their experimental detail and code was sufficient to only reproduce three of the graph datasets (3d, taxi, 3dd). Therefore, we develop two additional asymmetric graphs (traffic and octagon). The details of all the shortest-path graphs we use are provided in Table 9. Models were trained for 50 epochs at learning rate 5e-5 and 50 epochs at 5e-6.

## E.1  EXTENSIONS OF BREGMAN DIVERGENCE FOR THE TRIANGLE INEQUALITY

For the first we draw from mathematical literature demonstrating that metrics can be induced from the square root of certain symmetrized Bregman divergences, depending on constraints on $\phi$ Chen et al. (2008). We learn the square root of the Bregman divergence to provide a soft inductive bias (as an illustrative example, Euclidean distance is a metric but squared Euclidean distance is not).

For the second we introduce a modification of the Bregman divergence known as the Generalized Symmetrized Bregman divergence. As shown by Acharyya et al. (2013), the square root of such a

divergence is guaranteed to satisfy the triangle inequality. This divergence is defined as $D_\phi^{gsb}(x,y) = D_\phi(x,y) + D_\phi(y,x) + \frac{1}{2}\|x-y\|_2^2 + \frac{1}{2}\|\nabla\phi(x) - \nabla\phi(y)\|_2^2$.

There is an inherent tradeoff between the two extensions as Bregman-sqrt can be asymmetric but still does not require satisfying the triangle inequality, while Bregman-GS is symmetric but always satisfies the triangle inequality.

## F  ADDITIONAL HYPERPARAMETER DETAILS

We followed the hyperparameter specifications for the Deepnorm and Widenorm results as stated in Pitis et al. (2020). The Widenorm used 32 components with size 32, concave activation size 5, and max-average reduction. For the Deepnorm we used the neural metric version which gave the strongest performance in their paper: 3 layers with dimension 128, MaxReLU pairwise activations, concave activation size 5, and max-average reduction. We adapt their PyTorch code from https: //github.com/spitis/deepnorms. In the graph distance task, results show the same learning pattern (and relative performances between models) but the overall error magnitudes that we obtain are rather different than their reported results.

We re-implemented the Deep-div method following the description in Cilingir et al. (2020). The number of affine sub-networks stated in their paper varied but was generally set to low values such as 10, for the purpose of matching the number of classes in their classification tasks. In their appendix they experiment with increasing numbers of sub-networks and find best results at 50. For this reason we set 50 for our experiments. Following their paper, we use small FNNs for the max-affine components.

Our NBD uses a 2 hidden layer FICNN with width 128 for $\phi$. We found that our results are robust to the depth and width of the FICNN.

### F.1  ACTIVATION FOR FICNN

While developing theory using the softplus directly is harder because of the non-linearity, we believe that the approximation error using softplus activations is upper bounded by the ReLU approximation error. This is because the softplus can be made arbitrarily close to ReLU by making $\alpha$ large in the operation $g_\alpha(x) = \log(1 + \exp(\alpha x))/\alpha$. As the $\alpha$ value could be learned in the affine components of the neural network, a ICNN with ReLU can be closely expressed as an ICNN with softplus. So we can simply substitute $g_\alpha(x)$ for ReLU(x) in the original proofs, and therefore our optimal error is less than or equal to the ReLU error. (In particular when the true function has curvature then it will be strictly better than ReLU).

## G  OVERLAP DETAILS

We use the INRIA Holidays dataset Jegou et al. (2008), which contains over 800 high-quality vacation pictures contributed by the original authors. Many consecutive-numbered images (e.g. 129800.jpg, 129801.jpg, 129802.jpg) are retakes of the same scene, which would interfere with assigning zero overlap to different images. To address this we only use images ending in 00.jpg, which are all different scenes. This leaves 300 images, which we resize to 72x72 then apply a 64x64 center-crop. The training set consists of $10,000$ pairs sampled with random crops each epoch from the first 200 of the images, while the test set is a fixed set of $10,000$ pairs with crops drawn from the last 100. We set a 25% chance for a given pair to come from different images and receive a divergence of 1. All models were trained with batch size 128, Adam optimizer with learning rate 5e-4, and embedding dimension of 128. The embedding network consists of four 3x3 convolutional layers (32, 64, 128, 256 filters respectively) with 2x2 max pooling layers followed by two linear layers with hidden dimension 256.

We compute overlap as the percent of non-intersecting area from the crops. We show this in Fig. 5.

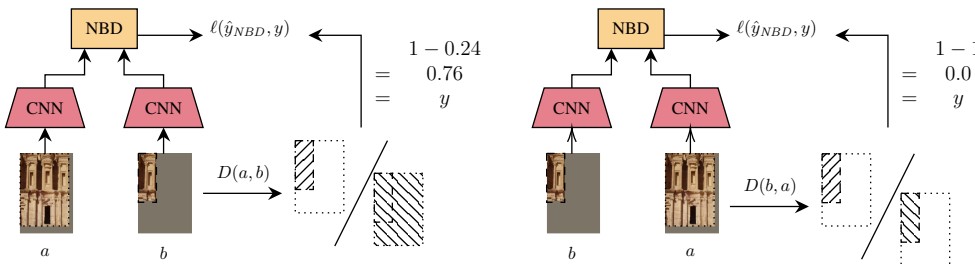

Figure 5: Demonstration of how the overlap distance is computed in our setup. Ground-truth distance is the intersection of the shared images divided by the area of the first image.

## H  BREGMAN RANKING AND CLUSTERING STANDARD DEVIATIONS

Full version of Table 3. Standard deviations in small font, means in regular font.

| Dataset | Model | MAP | AUC | Purity | Rand Index |
|---|---|---|---|---|---|
| abalone | Deep-div | $0.281_{0.01}$ | $0.645_{0.02}$ | $0.377_{0.02}$ | $0.660_{0.04}$ |
| | Euclidean | $0.301_{0.01}$ | $0.666_{0.01}$ | $0.422_{0.03}$ | $\mathbf{0.750}_{0.01}$ |
| | Mahalanobis | $0.310_{0.01}$ | $0.677_{0.01}$ | $0.419_{0.02}$ | $\mathbf{0.750}_{0.01}$ |
| | NBD | $\mathbf{0.316}_{0.01}$ | $\mathbf{0.682}_{0.01}$ | $\mathbf{0.432}_{0.03}$ | $\mathbf{0.750}_{0.01}$ |
| | PBDL | $0.307_{0.01}$ | $0.659_{0.01}$ | $0.386_{0.02}$ | $0.735_{0.02}$ |
| balance-scale | Deep-div | $0.804_{0.03}$ | $0.859_{0.02}$ | $0.869_{0.02}$ | $0.828_{0.03}$ |
| | Euclidean | $0.611_{0.01}$ | $0.666_{0.01}$ | $0.633_{0.06}$ | $0.568_{0.04}$ |
| | Mahalanobis | $0.822_{0.01}$ | $0.854_{0.01}$ | $0.851_{0.06}$ | $0.761_{0.05}$ |
| | NBD | $\mathbf{0.887}_{0.01}$ | $\mathbf{0.915}_{0.01}$ | $\mathbf{0.898}_{0.02}$ | $\mathbf{0.872}_{0.03}$ |
| | PBDL | $0.836_{0.02}$ | $0.855_{0.02}$ | $0.872_{0.02}$ | $0.814_{0.03}$ |
| car | Deep-div | $0.787_{0.01}$ | $0.757_{0.01}$ | $0.852_{0.04}$ | $0.750_{0.04}$ |
| | Euclidean | $0.681_{0.00}$ | $0.589_{0.00}$ | $0.704_{0.02}$ | $0.523_{0.03}$ |
| | Mahalanobis | $0.787_{0.01}$ | $0.752_{0.01}$ | $0.778_{0.02}$ | $0.654_{0.03}$ |
| | NBD | $\mathbf{0.820}_{0.01}$ | $\mathbf{0.803}_{0.01}$ | $\mathbf{0.860}_{0.01}$ | $\mathbf{0.758}_{0.02}$ |
| | PBDL | $0.798_{0.01}$ | $0.775_{0.01}$ | $0.854_{0.01}$ | $0.750_{0.02}$ |
| iris | Deep-div | $0.945_{0.03}$ | $0.967_{0.02}$ | $0.811_{0.16}$ | $0.820_{0.16}$ |
| | Euclidean | $0.827_{0.02}$ | $0.897_{0.01}$ | $0.820_{0.07}$ | $0.828_{0.05}$ |
| | Mahalanobis | $0.946_{0.03}$ | $0.973_{0.01}$ | $0.884_{0.12}$ | $0.879_{0.11}$ |
| | NBD | $\mathbf{0.957}_{0.02}$ | $\mathbf{0.977}_{0.01}$ | $\mathbf{0.909}_{0.10}$ | $\mathbf{0.902}_{0.10}$ |
| | PBDL | $0.943_{0.03}$ | $0.967_{0.02}$ | $0.889_{0.14}$ | $0.888_{0.13}$ |
| transfusion | Deep-div | $0.648_{0.01}$ | $0.525_{0.02}$ | $\mathbf{0.756}_{0.03}$ | $0.621_{0.04}$ |
| | Euclidean | $0.666_{0.01}$ | $0.536_{0.01}$ | $0.748_{0.03}$ | $0.563_{0.04}$ |
| | Mahalanobis | $0.680_{0.01}$ | $0.570_{0.01}$ | $0.750_{0.03}$ | $0.543_{0.05}$ |
| | NBD | $\mathbf{0.695}_{0.01}$ | $\mathbf{0.603}_{0.01}$ | $\mathbf{0.756}_{0.03}$ | $0.600_{0.04}$ |
| | PBDL | $0.637_{0.01}$ | $0.504_{0.01}$ | $0.748_{0.03}$ | $\mathbf{0.622}_{0.03}$ |
| wine | Deep-div | $\mathbf{0.983}_{0.02}$ | $\mathbf{0.987}_{0.01}$ | $0.953_{0.08}$ | $0.947_{0.08}$ |
| | Euclidean | $0.844_{0.02}$ | $0.884_{0.02}$ | $0.902_{0.07}$ | $0.887_{0.06}$ |
| | Mahalanobis | $0.949_{0.02}$ | $0.970_{0.01}$ | $0.944_{0.10}$ | $0.940_{0.09}$ |
| | NBD | $0.969_{0.02}$ | $0.980_{0.01}$ | $\mathbf{0.960}_{0.05}$ | $\mathbf{0.948}_{0.06}$ |
| | PBDL | $0.978_{0.02}$ | $0.982_{0.01}$ | $0.820_{0.14}$ | $0.823_{0.12}$ |

Table 10: Across several real datasets, a learned Bregman divergence is superior to Euclidean or Mahalanobis metrics for downstream ranking (MAP, AUC) and clustering (Purity, Rand Index) tasks. Furthermore, our approach NBD consistently outperforms the prior Bregman learning approaches, Deep-div and PBDL, on most datasets. MAP = mean average precision, AUC = area under curve

# I  FIGURE (BREGMNIST) INCLUDING SYMMETRIC CASE

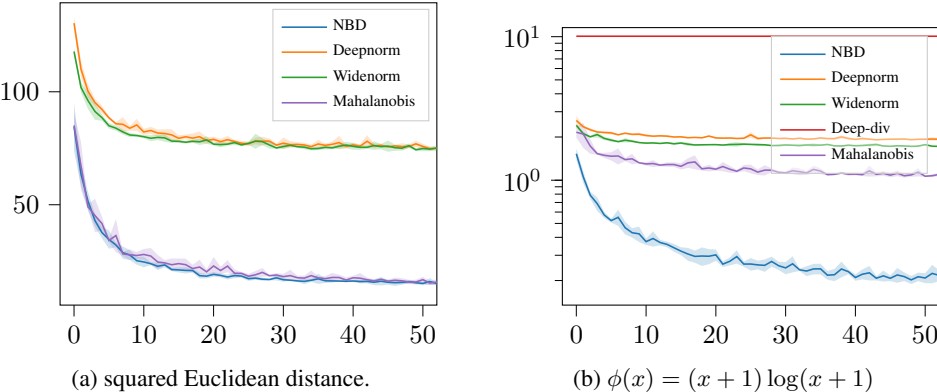

(a) squared Euclidean distance.

(b) $\phi(x) = (x+1)\log(x+1)$

Figure 6: MSE (y-axis) after epochs of training (x-axis), where NBD performs best in the symmetric (left) and asymmetric (center, right) Bregman learning tasks. We see Mahalanobis performs well in the symmetric task (left) since it is correctly specified for the ground truth, but performs relatively poorly in the asymmetric case (right). Deep-div is unable to learn effectively in either (not shown on left because the error was too high).

