# OpenReview forum: "Neural Bregman Divergences for Distance Learning"
_ICLR.cc/2023/Conference — ICLR 2023 poster_

### Official Review · Reviewer_GJoF · 2022-10-19

**Confidence:** 3
**Correctness:** 3
**Technical Novelty And Significance:** 2
**Empirical Novelty And Significance:** 2
**Recommendation:** 5

**Clarity, Quality, Novelty And Reproducibility:**

The writing is not very good.
- The end of Section 1 mixes the contributions and paper outline. Consequently, the major highlight of the work is unclear.
- The description of the main method is too short (Sections 2.2 and 2.3). The choice of ICNN is not well-motivated. Was it used for Bregman divergence learning before? Why is the ICNN parameterization better? A theoretical analysis is missing.


**Strength And Weaknesses:**

Strength:

The authors adopted a more generic method to parameterize the phi function, which seems better than several other existing Bregman divergence learning approaches in various settings.


Weakness:

I doubt the usefulness of non-Euclidean distance learning. The authors enumerate the use in retrieval, clustering, and ranking in the introduction. However, there is little evidence in the paper to show significant improvement in these applications compared to Euclidean approaches. Instead, the authors focused on various artificial settings of the pairwise distances. Winning in such artificial scenarios may not justify the new method.

Section 5.1 seems to be a realistic application. But the MAP@10 values are low compared with the state-of-the-art. Suppose we just use an image classifier (e.g., ResNet or Vision Transformer) and calculate the MAP@10 using the softmax outputs and their Euclidean distances. What is the benefit of using the proposed method?

It is infeasible to show that NBD works well for arbitrary non-Bregman distance learning. In Section 5.4, the authors have to modify the distance function as a remedy, which makes the claims looser.

The experiment setting is not clear. How many supervised pairs were used in training (compared to the number of data points)? Which pairs did you sample?

Table 4: most SVHN results are 96.9, which looks unnatural. Why?


**Summary Of The Paper:**

The paper presents a new method called NBD to learn Bregman divergences between data points using a deep neural network. The proposed method is tested on some synthetic and small machine learning benchmark data sets. The results show that NBD outperforms several other metric learning approaches for some tasks.

**Summary Of The Review:**

The paper seems to have a step forward in non-Euclidean distance learning. However, the main contribution is unclear, and the impact of the proposed method is questionable. I tend to give borderline rejection.

---

> ### Author Response · Authors · 2022-11-08
> **Reply to GJoF, detail questions**
>
> *Re usefulness of non-Euclidean distance learning*:
> As the reviewer notes, we highlight retrieval, ranking, clustering tasks as useful problems of interest. On these, our method in fact does show significant improvement over Euclidean approaches, including up to 45\% improvement on MAP and 37\% improvement in AUC (Table 2). Moreover, it is confirmed in Table 1 that Euclidean distance is highly sub-optimal at clustering non-Gaussian data. This is a clear demonstration of the advantage of non-Euclidean distance learning, as training a metric reflective of the actual data instantly raises clustering quality. Considering that many real datasets are not going to be Gaussian in nature, this is non-artificial evidence supporting non-Euclidean metric learning.
>
> We do construct artificial tasks with labeled divergences for regression in our paper. As discussed in response to rc5T, we intentionally designed them as an important validation of the underlying mathematical validity of our approach. In these inherently Bregman tasks, our method significantly outperforms the previous deep Bregman learning method (Deep-div).
>
> While the need for asymmetric learning may not exist for most practitioners today, we do understand that there are tasks where non-Euclidean distances are needed, and the community is showing a growing interest in them per our related work. As this research area develops and grows over time, it may become more broadly relevant --- but we argue our current results show a significant step forward over what is possible today.
>
> *Re Section 5.1*: The ultimate goal is to learn a metric from data that can provide a useful distance measure between any two samples. An effective metric learning approach is useful in many contexts where a softmax classifier simply would not work. In this task MAP@10 is used to assess the validity of the learned distance measure by using it to classify new samples by proxy. So while we could train a softmax classifier using SOTA architectures and likely achieve stronger classification accuracy, that is not the purpose of the experiment. Our comparisons are useful for understanding the relative performances of metric heads even if we do not beat the best classifiers in classification accuracy.
>
>
> *Re section 5.4*: We do not feel that modifying the Bregman divergence for triangle inequality-adhering tasks is a weakness of our approach. After all, it is a known property of the Bregman divergence that it does not adhere to the triangle inequality. Therefore, the modification is not a cop-out that dilutes the usefulness of our method. Rather, it is a mathematically sound adaptation to the particular constraints imposed by the graph distance task and highlights the flexibility of our framework to handle task-specific innovation. Consequently we feel that the modifications add to the novelty of our approach.
>
>
> *Re experiment setting:* In our labeled divergence experiments, pairs are generated on the fly and the divergence is computed between the pair as the target. For 4.2 there are infinite possible pairs (as the numbers are randomly sampled), while for 4.3, 5.2, 5.3, 5.4 there are $N^2$ possible pairs, where $N$ is the dataset size. The pairs are uniformly randomly sampled from data and we assign an epoch to be of length equal to $N$ pairs (so an epoch is the usual length, but not all data has necessarily been seen).
>
> In our triplet loss experiments (4.1, 4.2, 5.1), we iterate over batches of data in an epoch in the standard way. Then, given a data batch, we mine all triplets from the batch which give non-zero triplet loss. (Note that triplets which correctly satisfy the relative distance constraints do not contribute to the triplet loss.) So in a batch of size $|B|$ there are at most $O(|B|^3)$ triplets. This is a conventional technique to metric learning as used in Musgrave et al (2020), for example.
>
> The epoch and batch sizes themselves are provided in Appendix D. These are useful questions so we will describe these points in our paper to improve clarity to readers.
>
> *re Table 4:* We think it is because SVHN is simpler than the other datasets, so that the use of ResNet-18 as a base model is expressive enough to basically solve the tasks to near-oracle performance regardless of the metric head. Thus the models converge to the same endpoint. We will run an experiment with simpler architectures to check if our analysis is correct.
>
> **update re Table 4:**  Using a CNN with 4 conv layers (simpler than the ResNet) we found that on SVHN,
>
> | Metric | MAP@10                 |
> |:--------------:|---------------------|
> | NBD            | 92.5|
> | Deep-Div           | 22.4    |
> | Euclidean        | 92.5      |
> | Deepnorm       | 92.2        |
> | Widenorm       | 92.3        |
>
> which confirms that the models are similar (except Deep-div), but due to the *simpler base network*, are not able to converge to the *same* oracle performance. We hope this helps answer the reviewer's question.

---

> > ### Comment · Reviewer_GJoF · 2022-11-24
> > **The revision is better, but my main concern remains**
> >
> > The revision is better than the original submission. Most writing issues in my review have been solved.
> > But my major concern remains: there is no convincing real-world application.
> > * Table 1 is for cluster analysis, but all data sets are synthetic. Actually, there are many real-world benchmark datasets for cluster analysis. For example, the well-known MNIST dataset is ready for clustering comparison.
> > * Table 2 is for information retrieval, but all data sets are small and mainly used for **classification** instead of retrieval. A simple Google search can return many existing benchmark datasets for retrieval, e.g.,
> >   - https://paperswithcode.com/task/information-retrieval/latest
> >   - https://github.com/harpribot/awesome-information-retrieval#datasets
> >
> > By the current presentation, I cannot agree with the claim "Learning a task-relevant metric among samples is a common application of machine learning, with use in retrieval, clustering, and ranking."

---

> ### Author Response · Authors · 2022-11-08
> **Reply to GJoF, writing clarity**
>
> >The end of Section 1 mixes the contributions and paper outline. Consequently, the major highlight of the work is unclear.
>
> We propose the new set of paragraphs to address this issue and separate contributions from outline:
>
> Our NBD is the first non max-affine approach to learning a Bregman divergence. This avoids a limitation in the learning of max-affine models prior works suffer. Beyond learning difficulty, prior methods did not fully model the space of possible Bregman divergences, an issue NBD rectifies by construction via a key insight. Rather than attempt to model the divergence $D_\phi$ directly as prior works, we instead model the generating function $\phi(x)$, and then use $\phi(x)$ to implement the full divergence $D_\phi$. To demonstrate efficacy, we leverage prior and propose several new benchmarks of asymmetry organized into three types of information they provide: 1) quality of learning a Bregman divergence directly, 2) ability to learn a Bregman divergence and a feature extractor jointly, and 3) effectiveness in asymmetric tasks where the ground truth is known to be non-Bregman. This set of tests shows how NBD is far more efficacious in representing actual Bregman divergences than prior works, while simultaneously performing better in non-Bregman learning tasks.
>
> The rest of or paper is organized as follows. In §2 we show how to implement NBD using an Input Convex Neural Network with key related work, and further related work in §3. In §4 we demonstrate multiple Bregman learning tasks including regression, ranking, and clustering.  Then §5 studies the performance of our method on
> asymmetric tasks where the underlying metric is not known to be Bregman, to show more general utility where prior Bregman methods fail. Finally we will conclude in §6.
>
> >The description of the main method is too short (Sections 2.2 and 2.3). The choice of ICNN is not well-motivated. Was it used for Bregman divergence learning before? Why is the ICNN parameterization better? A theoretical analysis is missing.
>
>
> A Bregman divergence requires a convex generating function. Previous Bregman learning works have only considered piecewise linear convex functions. When the true distance measure is smooth this results in irreducible approximation error, and in fact the previous formulations are also rather inefficient to train. Our use of ICNN is natural as it is a convex neural network by construction. We further make it a smooth convex function of the inputs and show that it is (1) more efficient and (2) also more accurate than prior methods. We will add this and more explanation to section 2.
>
> We hope these address your concerns about the readability of our paper and the clarity of our contribution.

---

### Official Review · Reviewer_rc5T · 2022-10-21

**Confidence:** 3
**Correctness:** 3
**Technical Novelty And Significance:** 3
**Empirical Novelty And Significance:** 4
**Recommendation:** 8

**Clarity, Quality, Novelty And Reproducibility:**

- The paper is well-written and easy to follow.
- It cites and discuss relevant previous work properly, and the motivation of the work is clear.
- Introducing ICNNs to learning Bregman divergences is novel as far as I know.
- I think the paper and the supplementary material provide sufficient details for reproducing their experiments.
- One thing that is not very clear to me is what labels are given in each experiment.

Questions:
- In Eq. (3), is $\tilde{y}$ fixed or varied with $\theta$ when we take the gradient?
- In Section 5.1, is the triplet loss the same as that for PBDL described in Section 2.1?
- In Algorithm 1, line 8, what loss does it use?

**Strength And Weaknesses:**

# Strengths
- The approach of directly learning the convex function by an ICNN seems interesting. Related papers on ICNNs support the representation power of the model.
- The paper presents quite extensive experiments, and the proposed method shows excellent performance.

# Weaknesses
- I have a concern with Eq. (2). The paper says "by construction, the resulting neural network satisfies convexity," but I don't think this is the case. If, additionally, $g$ were a non-decreasing function as assumed in Amos et al. (2017), I would understand the claim. I understand that the proposed method uses $g(x) = \log(1 + \exp(x))$ which is nondecreasing, so this may not be a big problem. However, I wonder if theories by Chen et al. (2019) and Pitis et al. (2020) apply to this activation function.
- If I understand well, some experiments assume that divergence values are given as labels, which I believe is not very realistic in practice.

**Summary Of The Paper:**

This paper proposes a method for learning a Bregman divergence using neural networks. A Bregman divergence is the divergence defined using a convex function, and any convex function has its corresponding Bregman divergence. The proposed method learns the divergence by representing the convex function by an Input Convex Neural Network (ICNN) (Amos et al., 2017). The paper provides a series of experiments to confirm the superiority of the proposed method over the previous methods.

**Summary Of The Review:**

Overall, I like the paper and the authors make great contributions with many empirical results. I have a few concerns that I mentioned above, but I tend to accept the paper.

---

> ### Author Response · Authors · 2022-11-08
> **Response to rc5T**
>
> Re concern with Eq. (2): Yes, the reviewer is correct and we appreciate their careful attention. We meant to say that $g$ is 'convex and non-decreasing' and have added it into the text. As the reviewer notes, our method uses the $g(x)=\log(1+\exp(x))$ (softplus) function which is essentially a smoothed version of ReLU, and thus the ICNN is guaranteed to be convex.
>
> To give some context on applying the theory, proofs from prior works, e.g. Chen et al (2019), Siahkamari et al (2019), have directly focused on ReLU/other piecewise linear activations which fit neatly into longstanding theory in max-affine approximations. While developing theory using the softplus directly is harder because of the non-linearity, we believe that the approximation error using softplus activations is upper bounded by the ReLU approximation error. This is because the softplus can be made arbitrarily close to ReLU by making $\alpha$ large in the operation $g_\alpha(x)=\log(1 + \exp(\alpha x)) / \alpha$. As the $\alpha$ value could be learned in the affine components of the neural network, a ICNN with ReLU can be closely expressed as an ICNN with softplus. So we can simply substitute $g_\alpha(x)$ for ReLU(x) in the original proofs, and therefore our optimal error is less than or equal to the ReLU error. (In particular when the true function has curvature then it will be strictly better than ReLU).
>
> Re divergence labels: It is true that in practical problems there will not be a divergence label. However, we use them in the first phase of our experiments as a validation of the mathematical theory that our method can successfully learn underlying Bregman divergences. The other methods we compared against are not effective at this in comparison.
>
> Eq. (3): $\tilde y$ does vary with $\theta$. Specifically $\tilde y = f_\theta(y)$, the embedding of $y$ using base network $f_\theta$. Then $\phi$ is treated solely as a function of $\tilde y$ and so its gradient is evaluated at the point $\tilde y$ without regard to the earlier network $f_\theta$.
>
> Triplet loss: It is nearly the same as the printed equation, except the margin is 0.2 instead of 1. We follow common settings from Musgrave et al (2020) and will clarify the differences further in Appendix D. The triplets are also mined from each batch rather than being a pre-determined set as in PBDL. As other reviewers have asked about similar details, we will make sure to clearly specify these in our revision.
>
> Loss function: This depends on the task. In ranking and clustering (e.g. 4.1, 5.1) it is the triplet loss. In regression (4.2, 4.3, 5.2, 5.3, 5.4) it is squared error loss. We will clarify this further in our Experiment details.
>
> Clarity on labels: We will clarify the labels for each task in addition to the other details we have described in this response. We thank the reviewer for bringing these details to our attention.

---

### Official Review · Reviewer_DTa2 · 2022-10-26

**Confidence:** 4
**Correctness:** 4
**Technical Novelty And Significance:** 2
**Empirical Novelty And Significance:** 2
**Recommendation:** 3

**Clarity, Quality, Novelty And Reproducibility:**

The writing can significantly be improved.
The novelty of this paper is limited.

**Strength And Weaknesses:**

Strength:

This paper developed a new Neural Bregman Divergence, which jointly learns a  Bregman measure and a feature extracting neural network. It also demonstrates that the proposed method more faithfully learns divergences over a set of both new and previously studied tasks, including asymmetric regression, ranking, and clustering.

Weakness:

1. The novelty of this paper is limited. The proposed method basically follows the existing Bregman learning methods (Siahkamari et al., 2020; Cilingir et al.,2020), and only jointly learns a Bregman measure and a feature extracting neural network in the proposed method.

2. The experimental results of non-Bregman learning do not outperform SOTA methods. Does it mean the proposed method NBD only achieves better result on specific tasks where the underlying ground truth is from Bregman divergence? Will it restrict the application of NBD?


**Summary Of The Paper:**

This paper addresses the limitation of previous methods for Bregman divergence learning and proposes a solution named Neural Bregman Divergences (NBD) using Input Convex Neural Network (ICNN), which can be computed efficiently and gives finer resolution to the generating function. Experimental results verify the performance of the proposed method on many metric learning tasks.

**Summary Of The Review:**

This paper developed a new Neural Bregman Divergence, which jointly learns a Bregman measure and a feature extracting neural network.  It also demonstrates that the proposed method more faithfully learns divergences over a set of both new and previously studied tasks, including asymmetric regression, ranking, and clustering. Some experimental results verify the effectiveness of the proposed method. However, the novelty of this paper is limited.

---

> ### Author Response · Authors · 2022-11-08
> **Reply to DTa2 (part 1)**
>
> Re weakness 1: We believe we may not have sufficiently communicated the difference between our NBD and prior Siahkamari et al., 2020 (PBDL) &  Cilingir et al.,2020 (Deep-Div).
>
> Our method NBD is the first to directly optimize the generating function $\phi(x)$ as a means of implementing the fully general Bregman divergence $D_\phi(x, y) = \phi(x) - \phi(y) - \langle \nabla \phi(y), x - y \rangle$. In contrast, the prior works use a max-affine approach to tackle $D_\phi(x, y)$ directly. This causes scalability issues for both methods, as well as difficulty learning smooth functions and limits the scope of effective application.
>
> PBDL uses the max-affine approach so that they can find a solution that satisfies the Bregman properties as a linear program, since all parts of the max-affine representation of $\phi$ are linear. Deep-Div uses a similar max-affine $\phi$, expressed in a neural network, which also uses linearity to avoid dealing with the $- \langle \nabla \phi(y), x - y \rangle$ term from the definition of a standard Bregman Divergence.
>
> Our novelty comes in identifying that the whole class of Bregman divergence can be more generally represented by instead focusing on modeling the convex function $\phi(x)$ directly, and then using double-back propagation to implement the general equation of $D_\phi$ directly. This has never been done by any prior Bregman learning paper, and the tools to perform this approach have existed for longer than PBDL or Deep-Div being published. In this way our work is novel, as it is an approach never realized or conceptualized, and considerably different from all prior Bregman learning methods that are restricted to max-affine representations.
>
> As our Example 2.2 shows, the Deep-Div approach has a theoretical shortcoming in its ability to learn effectively. This is born out by our experiments where Deep-Div fails to learn representations in most cases, often performing worse than the Euclidean distance on asymmetric tasks that isolate the specific goal of learning a divergence or asymmetric measure. As noted in our paper, NBD outperforms the other Bregman methods in 41/44 cases.
>
> To help make this more clear in revision, we will include the below table in the paper/appendix.
>
> | Bregman Method | $D_\phi$ Representation                                                                  |            $\phi$ Representation           |  Learning Approach | Compute Complexity | Joint Learning |
> |:--------------:|------------------------------------------------------------------------------------------|:------------------------------------------:|:------------------:|--------------------|----------------|
> | NBD            | $D_\phi(x_i, x_j) = \phi(x_i) - \phi(x_j) - \langle \nabla \phi(x_j), x_i - x_j \rangle$ | $\phi(x)$ is a Input Convex Neural Network | Gradient Descent   | $O(\|\theta\|)$      | Yes            |
> | PBDL           | $D_\phi\left({x}_i, {x}_j\right)=z_i-z_j-{a}_j^T\left({x}_i-{x}_j\right)$                | $\phi(x_i) = \max_i (b_i^\top x + z_i)$    | Linear Programming | $O(n^3)$             | No             |
> | Deep-Div       | $D_\phi(x_i, x_j) = \phi_i(x_i) - \phi_i(x_j)$                                               | $\phi(x_i) = \max_i (b_i^\top x + z_i)$    | Gradient Descent   | $O(\|\theta\|+K)$    | Yes            |

---

> > ### Author Response · Authors · 2022-11-08
> > **Reply to DTa2 (part 2)**
> >
> >
> > Re weakness 2: We believe we have shown through our experiments on multiple prior and new tasks, that our NBD is the SotA for Bregman learning of asymmetric tasks. Our experiments cover multiple scenarios of which there are:
> >
> > 1. The underlying goal is to learn a Bregman measure for a ground truth Bregman task
> > 2. The underlying goal is to jointly learn a Bregman measure for an underlying Bregman ground truth and an embedding of the data from which the measure is computed.
> > 3. The underlying goal is jointly learn a Bregman measure when the ground truth is non-Bregman (i.e., imperfect specification of an asymmetric relationship)  with an embedding of the data for which the measure is computed.
> >
> > To the best of our knowledge no prior work has tackled all three cases, and our evaluation tackles both nuances of Bregman learning and type as well as shortcoming in prior tasks (i.e, the shortest path problem is better solved by forcing symmetry, which defeats the purpose). In all three cases we have compared against prior SOTA Bregman methods and obtains better results on 41/44 total comparisons against prior Bregman methods. When we compare against non-Bregman measures of asymmetry like Wide and Deep-norm, NBD is competitive performing best or second best in most cases, with Wide and Deep norm alternating in their performance when they do well. In particular, our tests where Wide or Deep norm performs best are non-Bregman, and have intrinsic benefits to one of those approaches making the comparison unfair to our method. For example, the shortest path task benefits from the triangle inequality that Bregman measures do not have, but Wide and Deep-norm do have. That we can make NBD competitive in this case shows that NBD has utility beyond purely known Bregman tasks.
> >
> > If the reviewer could help us by understanding what asymmetric task we missed from prior work, or prior Bregman asymmetric learning approach we missed - we are happy to try and quickly incorporate them into our tests. As far as we are aware, our work tests more tasks, and larger datasets, than  prior asymmetric learning paper has considered.
> >
> > >The writing can significantly be improved
> >
> > If the reviewer could help us by specifying what writing issues are present in the paper, we will earnestly endeavour to rectify these concerns.

---

### Official Review · Reviewer_XV53 · 2022-11-06

**Confidence:** 5
**Correctness:** 4
**Technical Novelty And Significance:** 4
**Empirical Novelty And Significance:** 4
**Recommendation:** 8

**Clarity, Quality, Novelty And Reproducibility:**

The overall quality of the writing is good. There are many sections in the paper which looks misleading at first but has a good flow in general.

**Strength And Weaknesses:**

The paper has many contributions and experiments. The idea of using input convex neural networks for learning Bergman divergences is new. The extensions to satisfy the triangle inequality and symmetry are also new. They have done extensive experimentation with different tasks and have compared to a significant number of competing methods. Further they have defined interesting divergence regression tasks such as BregMNIST and BregCifar.

**Summary Of The Paper:**

This paper provides a method for learning Bergman divergences using input convex neural networks as the convex generating function. This eliminates the shortcomings of previous work on the topic including non convexity of the neural network for deep divergence learning and high computational complexity of linear Bregman divergence learning. They further use CNN's as feature extractors and implement joint training. They experiment with various tasks such as regression, ranking and clustering and outperform the competing methods. Further they extend the Bregman divergence learning in two directions, one where the divergence satisfies both triangle inequality and symmetry and another where only triangle inequality is satisfied.

**Summary Of The Review:**

I think the paper has done enough work and has good quality for publication in the conference. Accept.

---

> ### Author Response · Authors · 2022-11-08
> **Reply to XV53**
>
> We appreciate the reviewer's evaluation, particularly the note about the breadth of our experiments and results. We worked hard to fit a large amount of content into the ICLR page limit, so it is appreciated and we are glad you seem to have enjoyed the paper. We will endeavor to improve the flow of the paper to avoid misleading sections, any specific details that you found misleading are appreciated so we can ensure your concern is fully addressed.

---

### Author Response · Authors · 2022-11-18
**Updated manuscript in response to reviewers**

We thank the reviewers for their helpful comments, questions, and feedback. We have incorporated their suggestions into the new version of our manuscript, with changes highlighted in blue. While space for change was limited due to the 9-page constraint, we managed to add information where possible up to the space limit. Of particular note:
- We made our Introduction clearer by separating our contributions from the paper outline. The new outline clarifies the separation between key related works (max-affine Bregman approaches) in Section 2 and other related works (other asymmetric learning) in Section 3.
- We clarify the experimental targets in figures and tables.
- We added more experimental detail to clarify our experiments. In particular, additional sections in Appendix D describing an overall experimental procedure for regression tasks with mean square error loss, and clustering/ranking tasks with triplet loss. Here we also define the meanings of batch and epoch sizes in our triplet learning experiments.
- More detail to Section 2 explaining our use of ICNN, with discussion of theoretical representation capacity of our softplus activation. Also emphasizes that our approach improves on both efficiency and representational accuracy over the max-affine baselines.

We believe these changes have improved the readability and clarity of our manuscript. We hope that our changes adequately address all the reviewers' concerns and thank them again for their time and attention.

---

### Decision · Program_Chairs · 2023-01-20

**Decision:**

Accept: poster

**Justification For Why Not Higher Score:**

Though the paper meets the bar for publication, its novelty is somewhat low.  Further, the method may not be broadly of interest.

**Justification For Why Not Lower Score:**

See above.  The positives outweigh the negatives in my opinion, and I have taken a close look at the reviews and the paper.

**Metareview: Summary, Strengths And Weaknesses:**

Thanks for your submission to ICLR.

This was a borderline paper, with two quite positive reviewers and two more negative reviewers.  On the positive side, the reviewers liked the use of input convex neural networks to the Bregman divergence learning problem, and appreciated the experimental results.  On the negative side, some reviewers questioned the novelty of the approach as well as the overall usefulness of learning non-Euclidean distances.

After reading through this paper myself, I tend to agree with the more positive reviewers on this paper.  In particular, I do think there is novelty in using ICNNs for the BD learning problem, and find that it yields a much more scalable and practical solution as compared to the methods described in Siahkamari et al. and Cilingir et al.  I am also not particularly concerned with the issue of usefulness---non-Euclidean distances have been shown to be useful in a variety of contexts---this is well established---and the authors of this paper need not further establish this.

**Note From Pc:**

if the above contains the word "oral" or "spotlight" please see: "oral" presentation means -> notable-top-5% and "spotlight" means -> notable-top-25%. As stated in our emails, we are disassociating presentation type from AC recommendations

**Summary Of Ac-Reviewer Meeting:**

Despite repeated emails, I did not get a response from all reviewers and so could not schedule a meeting.  I did get a chance to talk one-on-one with one of the reviewers to go over their thoughts more deeply on the paper.